# Differential Disrupting Effects of Prolonged Low-Dose Exposure to Dichlorodiphenyltrichloroethane on Androgen and Estrogen Production in Males

**DOI:** 10.3390/ijms22063155

**Published:** 2021-03-19

**Authors:** Nataliya V. Yaglova, Dibakhan A. Tsomartova, Sergey S. Obernikhin, Valentin V. Yaglov, Svetlana V. Nazimova, Elina S. Tsomartova, Elizaveta V. Chereshneva, Marina Y. Ivanova, Tatiana A. Lomanovskaya

**Affiliations:** 1Laboratory of Endocrine System Development, Federal State Budgetary Scientific Institution “Research Institute of Human Morphology”, 117418 Moscow, Russia; dtsomartova@mail.ru (D.A.T.); ober@mail.ru (S.S.O.); vyaglov@mail.ru (V.V.Y.); pimka60@list.ru (S.V.N.); tselso@yandex.ru (E.S.T.); 2Department of Histology, Cytology, and Embryology, Federal State Funded Educational Institution of Higher Education I.M. Sechenov First Moscow State Medical University, 119435 Moscow, Russia; yelizaveta.new@mail.ru (E.V.C.); ivanova_m_y@mail.ru (M.Y.I.); tatiana_80_80@inbox.ru (T.A.L.)

**Keywords:** DDT, low-dose exposure, endocrine disruptor, androgen, estrogen, progesterone, gonads

## Abstract

Dichlorodiphenyltrichloroethane (DDT) is the most widespread, persistent pollutant and endocrine disruptor on the planet. Although DDT has been found to block androgen receptors, the effects of its low-dose exposure in different periods of ontogeny on the male reproductive system remain unclear. We evaluate sex steroid hormone production in the pubertal period and after maturation in male Wistar rats exposed to low doses of o,p’-DDT, either during prenatal and postnatal development or postnatal development alone. Prenatally and postnatally exposed rats exhibit lower testosterone production and increased estradiol and estriol serum levels after maturation, associated with the delayed growth of gonads. Postnatally exposed rats demonstrate accelerated growth of gonads and higher testosterone production in the pubertal period. In contrast to the previous group, they do not present raised estradiol production. All of the exposed animals exhibit a reduced conversion of progesterone to 17OH-progesterone after sexual maturation, which indicates putative attenuation of sex steroid production. Thus, the study reveals age-dependent outcomes of low-dose exposure to DDT. Prenatal onset of exposure results in the later onset of androgen production and the enhanced conversion of androgens to estrogens after puberty, while postnatal exposure induces the earlier onset of androgen secretion.

## 1. Introduction

Low-dose nonoccupational exposure to endocrine-disrupting chemicals is a global problem due to their worldwide dissemination [1,2,3]. Endocrine-disrupting chemicals affect hormone production by binding to hormone receptors, interfering with cell signaling pathways, and changing the functional activity of hormone-producing cells [1,4]. Dichlorodiphenyltrichloroethane (DDT) and its metabolites are the best known and most widespread endocrine disruptors on the planet [5,6]. DDT was extensively used in the 20th century as an insecticide in agriculture and public health until its ban in the 1970s following the Stockholm Convention on Persistent Organic Pollutants. Despite the evident toxic and carcinogenic effects of high-dose exposure, the World Health Organization later recommended the reintroduction of DDT for vector-borne disease control, as it is one of the most effective insecticides [7]. This is the background for the continuous dissemination of DDT in the environment. Its long half-life and ability to accumulate in the food chain ensures the low-dose persistence of DDT in all ecosystems of the planet and its resulting negative impact on human health [7,8,9,10,11]. The main route of exposure to DDT is the ingestion of contaminated food products [12,13]. Screening studies initiated by the World Health Organization and the United Nations Environment Program show that detectable levels of DDT and its metabolites are found in almost 100% of the population [14,15,16]. DDT, due to its low molecular weight and high lipophilicity (see Appendix A), easily penetrates the histohematogenous barriers and accumulates in various cells, especially those with a high lipid content in the cytoplasm, like steroid-producing cells [17]. Studies have revealed an antiandrogenic effect of DDT mediated by the blockage of androgen receptors and impaired male fertility after high-dose exposure [18,19,20,21]. The disrupting effects of DDT on the synthesis of androgens, as well as the metabolism and reception of estrogens, especially in males, have been studied to a lesser extent. Investigations in the field of endocrine disruption are complicated primarily due to the fact that the standards for toxicological studies are not applicable to endocrine-disrupting chemicals. There are no safe doses for endocrine disruptors, which leads researchers to study extremely low levels of exposure, similar to physiological doses of endogenous hormones [17]. An attempt to increase the dose leads to the development of toxic manifestations and does not allow disruptor effects to be established. The persistence of low doses of DDT requires thorough study since the exposure begins in the prenatal period. An increased incidence of developmental, morphological, and functional abnormalities of the reproductive system and neoplastic processes of the female and male reproductive systems, as registered in infants of DDT-exposed mothers, prove that prenatal low-dose exposure produces a wide range of negative outcomes during prenatal and postnatal development [17,22,23,24,25,26,27]. In our previous studies, we revealed disorders in the synthesis of thyroid hormones and mineralo- and gluco-corticoids in rats exposed to low doses of DDT [28,29]. However, today, it is clear that studies on endocrine disruptors require an ontogenetic approach since their effect begins in the prenatal period. The endocrine disruptor may change the developmental program and affect both prenatal and postnatal histogenesis as a result. It is well-known that the expression of steroidogenic enzymes and the synthesis of sex steroids is initiated in the embryonic period; that is why elucidation of the time dependence of the outcomes of exposure is essential for the assessment of sexual maturation parameters and the risks of possible reproductive and oncologic disorders.

In the present study, we investigate the production of sex steroid hormones during puberty and after maturation in rats exposed to low doses of DDT, both during prenatal and postnatal development and during postnatal development alone, to differentiate the outcomes of prenatal and postnatal exposure.

## 2. Results

### 2.1. Changes in Gonadal Development

The examination of gonads did not reveal cryptorchidism or other evident anatomical abnormalities in DDT-exposed rats. In puberty, the relative gonad weight in rats that were both prenatally and postnatally exposed did not differ from the control, but in solely postnatally exposed rats, it significantly exceeded the control values (Figure 1). Relative gonad weight did not change with age in the control rats. Unlike the control, all DDT-exposed rats demonstrated an age-dependent decrease in this parameter. After puberty, the minimal values of gonad weight were in rats that were both prenatally and postnatally exposed (Figure 1).

### 2.2. Changes in Sex Steroid Precursors’ Secretion

The rats exposed to low doses of DDT during prenatal and postnatal development demonstrated higher serum levels of progesterone and 17OH-progesterone in the pubertal period compared to the control. Unlike the prenatally and postnatally exposed rats, the rats solely exposed to low doses of DDT in the postnatal period exhibited a lower level of progesterone; their 17-OH-progesterone was in the normal range (Figure 2). 

After sexual maturation, the profile of sex steroid precursors for the control rats presented decreased progesterone but elevated 17OH-progesterone levels. Prenatally and postnatally exposed rats demonstrated a reduction of both progesterone and 17OH-progesterone production after puberty. Postnatally exposed rats, on the other hand, showed a more pronounced decrease in progesterone and 17OH-progesterone production (Figure 2).

### 2.3. Changes in Androgen Hormones’ Production 

The testosterone and androstenedione concentrations in prenatally and postnatally exposed rats were significantly lower in puberty. In postnatally exposed rats, on the other than, the concentrations of androgens significantly exceeded the control values (Figure 3). 

After sexual maturation, the testosterone serum concentration increased twice in the control rats (Figure 3A), while their androstenedione levels significantly lowered (Figure 3B). Prenatally and postnatally exposed rats demonstrated a 10-fold increase in testosterone and a 1.5-fold reduction in androstenedione production with age. Nevertheless, both androgens presented significantly lowered levels after puberty compared to the control. Unlike the control rats, the postnatally exposed rats showed no changes in testosterone production with age; their testosterone levels after puberty were within the control range. Their androstenedione levels, meanwhile, reduced after puberty but were also similar to the control values (Figure 3). 

### 2.4. Changes in Estrogen Hormones’ Production 

The patterns of estrogen production in DDT-exposed rats during puberty also differed from those of the control rats. In prenatally and postnatally exposed rats, their serum estradiol was dramatically decreased and their estrone production was also attenuated, but the estriol serum content exceeded the control value by 6.7 times. In postnatally exposed rats, increases in the levels of estradiol and estrone were registered, while estriol production was similar to that of the control (Figure 4).

After sexual maturation, the serum levels of estradiol and estriol in the control rats did not change, though the estrone concentration was significantly lower when compared to during puberty (Figure 4). Age-dependent changes in estrogen production in the prenatally and postnatally exposed rats differed from those of the control rats. No changes in estrone levels were found, though the estriol production reduced and the estradiol serum content significantly increased. As such, prenatally and postnatally exposed rats had a lowered testosterone/estradiol ratio after puberty compared to the control (Figure 4). Age-dependent changes in the production of estrogens in postnatally exposed rats differed both from those of the control and prenatally and postnatally exposed rats. Estradiol levels significantly decreased with age and were lower than those of the control and prenatally and postnatally exposed rats. The serum levels of estrone, also decreased after sexual maturation, did not differ from the control values. The estriol concentration increased and exceeded the control values. Postnatally exposed rats demonstrated the highest testosterone/estradiol ratio (Figure 4).

## 3. Discussion

The present investigation revealed significant changes in the growth of male gonads and sex steroid production, which were induced by prolonged exposure to low doses of DDT. Differences in the above-mentioned abnormalities were noted, depending on the period when exposure began. Postnatal exposure promoted the growth of gonads during puberty, unlike prenatal and postnatal exposure, which slowed gonad development. This fact suggests implications of the diverse molecular mechanisms of growth control. Independently of the onset of exposure, low doses of DDT provoked the depletion of gonads after puberty. 

Hormone assays also revealed different outcomes of prenatal and postnatal onset of exposure. Prenatal onset of exposure resulted in raised levels of progesterone and 17OH-progesterone. Progesterone is known to be a source of the synthesis of adrenal and gonadal steroid hormones [30]. Oxidation of the C17 atom in a molecule of progesterone is an initial step of sex steroid production in rats since the lowered activity of 11β-hydroxylase in adrenals prevents the conversion of 17OH-progesterone to cortisol, making this intermediate a principal precursor of androgens and estrogens in rats [31,32,33]. Attenuated production of testosterone and androstenedione indicated disrupted 17OH-progesterone conversion to androgens, both in the gonads and adrenals. Our previous studies, which revealed delayed development of adrenal zona reticularis in prenatally and postnatally DDT-exposed rats, are in accordance with the present findings. Taken together, these demonstrate the inhibition of the proliferation and differentiation of sex steroid-producing adrenal cells as a morphogenetic mechanism of endocrine disruption [34]. 

An additional manifestation of disruption was found to be associated with estradiol synthesis. Extremely low levels of estradiol, as found in pubertal rats, suggest suppressed aromatase activity in peripheral tissues since extra-gonadal sites of estrogen synthesis are considered to be the major source of circulating estrogens [34,35,36]. However, in parallel with a decrease in the production of testosterone, estradiol, and estrone, a significant increase in the synthesis of estriol was revealed. Estriol is a product of estradiol and estrone hydroxylation, which occurs to a greater extent in the liver than the gonads [37]. It is likely that the observed increase in estriol synthesis is a compensatory phenomenon aimed at increasing the production of sex steroids since estriol, being a weak estrogen, has a dual function of acting as both an agonist and an antagonist of estrogen receptors [38].

After puberty, the production of sex hormones in developmentally exposed rats also differed from that of the control rats. Increased production of the most active estrogen, estradiol, was noted. Estradiol levels were similar to those found in pubertal control rats. Estradiol is known to be the product of testosterone aromatization or estrone reduction. The revealed decrease in the production of testosterone and estrone, and the correspondence of a decrease in their concentration to an increase in that of estradiol, indicate their conversion into estradiol as the most probable mechanism of its hyperproduction. Despite the decrease in estriol synthesis with age, its serum concentration exceeded the control values. This indicates that the disruptor effect of DDT is not only associated with competition with testosterone for the opportunity to bind to androgen receptors and impair receptor signaling in target cells, but it also implicates an increase in estrogen synthesis. The revealed alterations in sex hormone production suggest reproductive and somatic disorders in later life since the hyperproduction of estrogens and an unbalanced testosterone/estradiol ratio are associated with an increased risk of feminization and are also known to trigger metabolic disorders as well as estrogen-related cancers and cardiovascular diseases [39,40,41,42,43,44,45,46,47]. 

The disruptor effect on the synthesis of sex steroids was found to be distinct in rats exposed to low doses of DDT during different periods of ontogeny. A higher testosterone concentration, along with a decreased concentration of progesterone but normal levels of 17OH-progesterone, are indicative of the early onset of the gonadarche in postnatally exposed rats. An increased level of estrone secretion denotes the normal functioning of the adrenal zona reticularis and the timely onset of adrenarche. The most important feature of the postnatally exposed rats was a similar-to-control concentration of estradiol, with a two-fold increase in testosterone levels in puberty. These differences indicate the absence of increased conversion of androgens into estrogens, revealed after gonadarche in rats exposed to low doses of DDT during prenatal and postnatal periods. The non-increasing conversion of estradiol and estrone into estriol confirms the compensatory nature of the increase in its production at a similar age in rats with prenatal and postnatal exposure. 

After puberty, the rats exposed to low doses of DDT only in their postnatal development also presented no testosterone deficiency, nor an excess of estrogens. On the contrary, there was a decrease in estradiol production and, accordingly, an increase in estriol synthesis. As such, an increase in the conversion of androgens to estrogens was not observed either in puberty or in the postpubertal period. However, more adequate androgen and estrogen synthesis was associated with a sharp decrease in the secretion of progesterone and 17OH-progesterone. A combined decrease in the levels of steroid hormone precursors, associated with decreased relative gonad weight, indicates the depletion of steroidogenesis and suggests a further decrease in sex steroid production. A comparison of the revealed abnormalities in rats exposed to an endocrine disruptor at different periods of ontogeny shows that the changes are not only related to the duration of exposure. In fact, the effect on the developing adrenal glands and testes, in which steroidogenesis and androgen receptor expression begin even in the prenatal period [48,49,50], cannot be considered as a consequence of a longer term of exposure or to be a dose-dependent effect. Endocrine disruptors are known to have a non-linear mode of action [51], and the opposite effects of high and low doses of endocrine disruptors on the onset of puberty in males have been already reported [52,53]. The present findings and our previously obtained data indicate that changes in the initiation of androgen and estrogen synthesis are dependent on the time when exposure to the endocrine disruptor begins. The heterogeneity of disruptor effects during the sexual maturation of the organism requires the close attention of researchers since it can promote various pathologies in adulthood. 

## 4. Materials and Methods

### 4.1. Animals

Female Wistar rats aged 19–20 weeks old (*n* = 30) were obtained from the Scientific Center of Biomedical Technologies of the Federal Medical and Biological Agency of Russia. The animals were housed at +22–23 °C and given a pelleted standard chow ad libitum. 

### 4.2. Chemicals

o,p’-DDT, purchased from “Sigma-Aldrich” (USA), was used since it has the highest solubility in water compared to other isomers of DDT [54]. The purity was 99.5%. For the main characteristic of the substance see Appendix A. DDT was dissolved in tap water to the final concentration of 20 μg/L required for low-dose exposure. Tap water and rat pelleted chow were preliminarily tested for an absence of DDT and its metabolites by high-performance liquid chromatography and mass spectrometry in the Federal Budgetary Institution of Public Health.

### 4.3. Experimental Design

The female rats were randomized into three groups. The first group (*n* = 10) included intact dams, which received tap water during pregnancy and lactation. The dams of the second group (*n* = 10) received a solution of o,p’-DDT ad libitum, instead of tap water, after mating during pregnancy and lactation. The third group (*n* = 10) received the same solution of o,p’-DDT during lactation (*n* = 10). After weaning, the progeny of the DDT-exposed dams received the same solution of o,p’-DDT during their postnatal development. The main experimental group included the male progeny (*n* = 20) of the dams from the second group, which were exposed to low doses of o,p’-DDT prenatally and postnatally (PPE DDT group). The male progeny of the third group (*n* = 20), exposed to DDT only during their postnatal development (PE DDT group), was included in the experiment to differentiate the effects of prenatal exposure. The male offspring of intact dams (*n* = 32) were used as a control. The sample size of the control group was increased to assess possible fluctuations in the steroid profile of rats, associated with the formation of social ranking, that affect the secretion of sex hormones. Half of the control and exposed male rats were sacrificed at the 42nd day of postnatal development, which corresponds to the pubertal period after adrenarche (21st day) and prior to gonadarche (50th day) [33]. Other rats were sacrificed on the 70th day of postnatal development, which corresponds to the onset of the reproductive period. The rats were sacrificed at 9–10 a.m. by means of zoletil overdose. The average daily DDT consumption by the pregnant dams was 2.70 ± 0.19 μg/kg, by the lactating dams 2.47 ± 0.11 μg/kg, and by the offspring 3.30 ± 0.14 μg/kg. The received doses corresponded to the rates of daily dietary exposure of humans to DDT [6]. The investigation was performed in accordance with the handling standards and rules of laboratory animals. It was consistent with the “International Guidelines for Biomedical Research with Animals” (1985), the laboratory routine standards in the Russian Federation (Order of Ministry of Healthcare of the Russian Federation dated 19 June 2003 No.267), the “Animal Cruelty Protection Act” dated 1 December 1999, and the regulations of experimental animal operation as approved by the Order of the Ministry of Healthcare for the USSR No.577 dated 12 August 1977. All animal procedures were approved by the Ethics Committee of the Research Institute of Human Morphology (protocol N8a).

### 4.4. Determination of Gonad Weight

The total body mass of the anesthetized rats was measured. After surgical removal, the gonads were measured separately. The average gonad weight was calculated for every rat enrolled in the investigation. Their relative gonadal weight was calculated and expressed as the percent of body weight. 

### 4.5. Hormone Assays

The collected blood samples were incubated at room temperature for 30 min. Clotted blood was centrifuged at 1500 rounds per minute for 15 min. The serum was transferred into polypropylene tubes. The intermediates in sex steroid synthesis—progesterone and 17OH-progesterone; gonadal and adrenal androgens; total testosterone and androstenedione; and the estrogens estradiol, estrone, and estriol—were measured using an enzyme-linked immunosorbent assay according to manufacturer’s protocols (Cusabio, China, Biovendor, RayBiotech) with the “Anthos 2010” microplate reader at 450 nm.

### 4.6. Statistical Analysis

Statistical analyses were carried out using the software package Statistica 7.0 (StatSoft, Tulsa, OK, USA). The central tendency and dispersion of quantitative traits with an approximately normal distribution were presented as the mean and standard error of the mean (M ± SEM). Quantitative comparisons of independent groups of the same age were performed using ANOVA and the Duncan test for post hoc comparison. Age-dependent changes in each group were analyzed with Student’s *t*-test, taking into account the values of Levene’s test for the equality of variances. Differences were considered statistically significant at *p* < 0.05.

## 5. Conclusions

Low-dose exposure to DDT disrupts sex steroid production and impairs the initiation of androgen synthesis. The prenatal onset of exposure provokes a later activation of androgen production, suggesting impaired adrenarche, while postnatal exposure induces an earlier onset of androgen secretion. The enhanced conversion of androgens to estrogens after puberty and a significantly changed blood androgen/estrogen ratio were found to be the outcomes of prenatal, but not postnatal, exposure to DDT. Low-dose exposure to DDT also diminishes the C17-hydroxylation of progesterone, independently of the period when it begins, and may result in attenuated sex steroid production later in life.

## Figures and Tables

**Figure 1 ijms-22-03155-f001:**
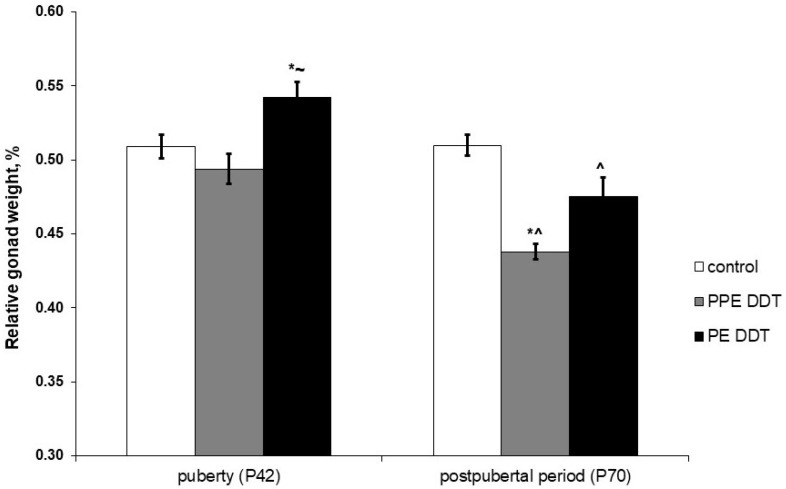
Effect of prolonged low-dose exposure to o,p’-dichlorodiphenyltrichloroethane (DDT), during different periods of ontogeny, on relative gonad weight in pubertal and postpubertal rats. Data are shown as mean ± SEM. P, day of postnatal development; *p* < 0.05 compared to control (*), compared to the prenatal and postnatal exposure (PPE) DDT group (~), and compared to pubertal period (^).

**Figure 2 ijms-22-03155-f002:**
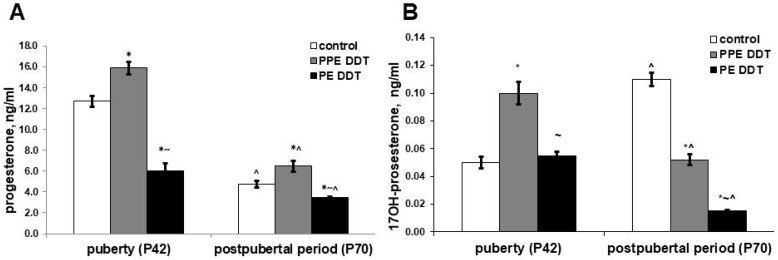
Changes in serum levels for sex steroid precursors in pubertal and postpubertal rats after prolonged low-dose exposure to o,p’-DDT during different periods of ontogeny. (**A**) Progesterone; (**B**) 17OH-progesterone. Data are shown as mean ± SEM. P, day of postnatal development; *p* < 0.05 compared to control (*), compared to the PPE DDT group (~), and compared to pubertal period (^).

**Figure 3 ijms-22-03155-f003:**
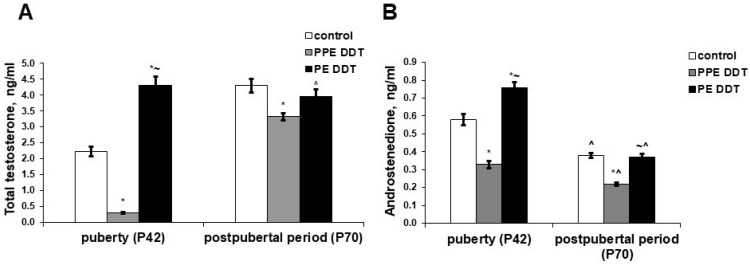
Changes in androgens’ serum levels in pubertal and postpubertal rats after prolonged low-dose exposure to o,p’-DDT during different periods of ontogeny. (**A**) Total testosterone; (**B**) androstenedione. Data are shown as mean ± SEM. P, day of postnatal development; *p* < 0.05 compared to control (*), compared to the PPE DDT group (~), and compared to pubertal period (^).

**Figure 4 ijms-22-03155-f004:**
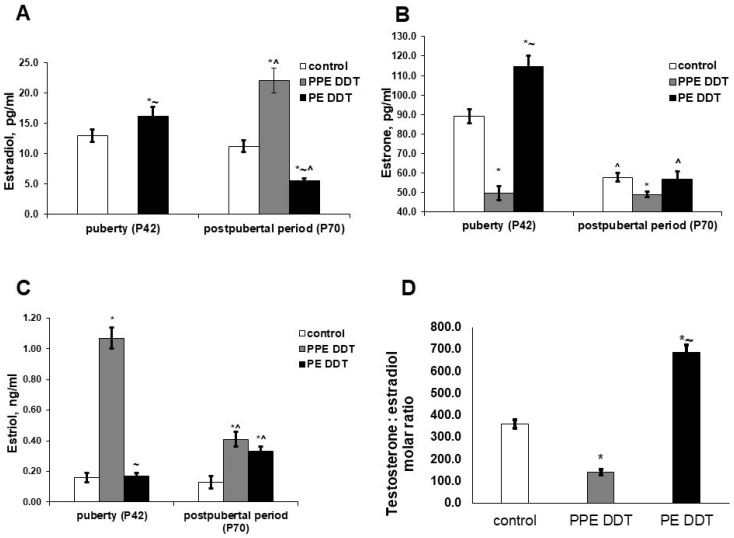
Changes in estrogen serum levels in pubertal and postpubertal rats after prolonged low-dose exposure to o,p’-DDT during different periods of ontogeny. (**A**) Estradiol; (**B**) estrone; (**C**) estriol; (**D**) testosterone/estradiol serum ratio in postpubertal rats at 70th day of postnatal development. Data are shown as mean ± SEM. P, day of postnatal development; *p* < 0.05 compared to control (*), compared to the PPE DDT group (~), and compared to pubertal period (^).

## Data Availability

The data presented in this study are available from the corresponding author upon reasonable request.

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
