# Peer review of "Differential Disrupting Effects of Prolonged Low-Dose Exposure to Dichlorodiphenyltrichloroethane on Androgen and Estrogen Production in Males"

_ijms, 2021, doi:10.3390/ijms22063155_

Round 1

Reviewer 1 Report

Manuscript „Differential disrupting effects of prolonged low-dose exposure 3 to dichlorodiphenyltrichloroethane on androgen and estrogen production“ is excellent paper which gives insight in crucial mechanism of endocrine disrupting agents. Iti s well written, informative and inovative.

Minor comments

For affiliation 1 city and country should be added. Corresponding author's name should be added

In title it should be added „in males“

Abstract

Word „ambiguous“ should be replaced with „age related“

Introduction

Line 33 Word „recent“ should be deleted as this is known for decades.

Results In Figure 2 it is not included (A) and (B).

Author Response

RESPONSE TO REVIEWER 1 COMMENTS

Point 1: For affiliation 1 city and country should be added.

Response 1: We added the required information.

Point 2: In title it should be added “in males”

Response 2: We agree with the comment and added “in males”

Point 3: Abstract. Word “ambiguous” should be replaced with “age-related”.

Response 3: We replaced “ambiguous” with “age-related”.

Point 4: Introduction, line 33. Word “recent” should be deleted as this is known for decades.

Response 5: We agree with the comment and modified the sentence as follows: “Endocrine disrupting chemicals affect hormone production by binding to hormone receptors, interfering with cell signaling pathways as well as changing functional activity of hormone-producing cells [1,4].”

Point 5: Results in Fig. 2 is not included (A) and (B).

Response 5: We checked the manuscript and found that Fig. 1 was inserted instead twice into the text. We deleted it and inserted the correct Figure 2.

The authors are very grateful to the reviewer for the work on the article.

Reviewer 2 Report

The study was designed to investigate exposure to DDT at critical life stages on androgen and estrogen production.   Further Improvement is required, it contained so many errors and was very hard to follow. To list a few:

  1. The age of female dams was not disclosed (line 233).
  2. The experiment design was very hard to follow. The authors might want to provide a flow chart to delineate how animals were grouped (lines 244-266).
  3. What was the post hoc test used for ANOVA? It is unclear which “control” group was used when comparison was conducted (controls from pre- and postnatal exposure or controls from postnatal exposure”?)
  4. No histology analysis was conducted on gonads.
  5. Figure 2 was not for progesterone.
  6. Line 258, it is unclear why the daily intake of DDT in lactating dams and offspring were negative values?
  7. Lines 131-132, the authors stated that “in postnatally exposed rats raise in levels of estradiol and estrone was registered. Estriol production was suppressed (Fig 4)” I assume the authors were referring to the levels of hormones in puberty in pre-and postnatal exposure group. However, figure 4 C indicated that estrone levels in PE DDT group was similar to controls.  Only reporting “suppression” of estrone compared to the levels in PPE DDT group is misleading.  

Author Response

RESPONSE TO REVIEWER 2 COMMENTS

Point 1: The age of female dams was not disclosed.

Response 1: We added the age (19-20 weeks) of the female dams to the text.

Point 2: The experiment design was very hard to follow. The authors might want to provide a flow chart to delineate how animals were grouped.

Response 2: We rewrote the experimental design and made it more consistent and clearer.

Point 3: What was the post hog test used for ANOVA? It is unclear which “control” group was used when comparison was conducted (controls for prenatal and postnatal exposure or controls from postnatal exposure?)

Response 3: The investigation included 3 groups (1- prenatal and postnatal exposure, 2 – postnatal exposure, 3 – control). Each group was subdivided into 2 subgroups (of pubertal and postpubertal age). ANOVA was used to compare the three groups of the same age. Duncan test was used for post hog comparison. (We added it into the description of statistical analysis). Age-dependent changes in control and exposed rats were analyzed with Student’s-test.

Point 4: No histology analysis was conducted on gonads.

Response 4: We carried out a visual examination and determination of the mass of the gonads in order to exclude anatomical abnormalities and to assess the rate of their development. Initially the study was not aimed at histological examination of the gonads. Our investigation revealed significant changes in androgen and estrogen production, which require thorough morphological study of gonads and adrenals. Since histological examination of gonads gives more information on spermatogenesis than steroidogenesis, proper investigation of Leidig cells function requires immunohistochemistry and electron microscopy. Unfortunately, our study has financial limitations, and we will be able to conduct morphological examination of the gonads only after receiving additional funding.

Point 5: Fig. 2 was not for progesterone.

Response 5: We are sorry. We inserted another figure by mistake. Now we have fixed the error and inserted the correct Fig. 2.

Point 6: Line 258, it is unclear why the daily intake of DDT in lactating dams and offspring were negative values.

Response 6:  The values of DDT intake are positive. The sign of dash was deleted.

Point 7: Lines 131-132, the authors stated that “in postnatally exposed rats raise in levels of estradiol and estrone was registered. Estriol production was suppressed (Fig 4)” I assume the authors were referring to the levels of hormones in puberty in pre-and postnatal exposure group. However, figure 4 C indicated that estrone levels in PE DDT group was similar to controls.  Only reporting “suppression” of estrone compared to the levels in PPE DDT group is misleading. Response 7: We agree that the sentence “Estriol production was suppressed” is misleading. We used it as follows “Estriol production was similar to control”. We also mentioned above that it refers to pubertal age.

The authors express their gratitude to the Reviewer for the comments and attention to the article.

Round 2

Reviewer 2 Report

Although the revised version has been improved.  There are still two points need to be addressed:  

  1. Why there were 32 male offspring were used in the control group while the other two groups used 20 male offspring?  Wistar rats typically have a large litter size.
  2. on lines 251-252, the authors stated "After weaning the progeny of the DDT-exposed dams received the same solution of o,p-DDT during postnatal development. " did these statement applied to all the offspring in both groups 2 and 3? In other words, the author had one group of offspring that had in utero and lactation exposure plus post weaning exposure until PND 42 or PND 70.  The second group had lactation exposure plus post weaning exposure until PND 42 or PND 70.   If this was true,  another group of offspring should be included in which the offspring  should only have DDT exposure until weaning.  These offspring should not have DDT exposure post-weaning.  The authors need to clarify exactly how the offspring in each group were treated.      

Author Response

RESPONSE TO REVIEWER 2 COMMENTS

Point 1: Why there were 32 male offspring were used in the control group while the other two groups used 20 male offspring?  Wistar rats typically have a large litter size.

Response 1: The number of rats in the experimental and control groups was determined not by the litter size, but by the need to obtain reliable data. It is always preferable to have a large control group in endocrine function studies. Rats are social animals, and sex steroid hormones production depends on social ranking. Since we investigated period of sexual maturation, when social ranking began to establish, we deliberately increased the number of control animals to determine the range of fluctuations in sex hormone concentrations.

Point 2: on lines 251-252, the authors stated "After weaning the progeny of the DDT-exposed dams received the same solution of o,p-DDT during postnatal development. " did these statement applied to all the offspring in both groups 2 and 3? In other words, the author had one group of offspring that had in utero and lactation exposure plus post weaning exposure until PND 42 or PND 70.  The second group had lactation exposure plus post weaning exposure until PND 42 or PND 70.   If this was true,  another group of offspring should be included in which the offspring  should only have DDT exposure until weaning.  These offspring should not have DDT exposure post-weaning.  The authors need to clarify exactly how the offspring in each group were treated.     

 Response 2: All the offspring in both groups 2 and 3 received DDT after weaning. Another group, exposed to DDT until weaning is nor required in this case, since the aim of the study was to differentiate the outcomes of prenatal exposure. That is why we have formed two groups with prenatal and postnatal exposure (group PPE DDT) and only postnatal exposure (group PE DDT).

The authors are very grateful to the reviewer for the work on the article.

Round 3

Reviewer 2 Report

The authors need to add the justification with respect to how the sample size was determined in the method part.  I have no further comments 

Author Response

We added the sentence "The larger sample size of the control group is due to the need to assess the steroid profile of rats, taking into account the formation in the period of maturation of social relations that affect the secretion of sex hormones." in the method part (lines 257-260).
